# Radiolysis-Assisted Direct Growth of Gold-Based Electrocatalysts for Glycerol Oxidation

**DOI:** 10.3390/nano13111713

**Published:** 2023-05-23

**Authors:** Nazym Tuleushova, Aisara Amanova, Ibrahim Abdellah, Mireille Benoit, Hynd Remita, David Cornu, Yaovi Holade, Sophie Tingry

**Affiliations:** 1Institut Européen des Membranes, IEM UMR 5635, University Montpellier, ENSCM, CNRS, 34090 Montpellier, France; naztuleushova@gmail.com (N.T.); david.cornu@umontpellier.fr (D.C.); 2Institut de Chimie Physique, UMR 8000-CNRS, Université Paris-Saclay, 91405 Orsay Cedex, France; aisara.amanova@universite-paris-saclay.fr (A.A.); i_abed2000@yahoo.fr (I.A.); mireille.benoit@universite-paris-saclay.fr (M.B.); hynd.remita@u-psud.fr (H.R.)

**Keywords:** gold and silver particles, radiolysis, glycerol electro-oxidation, carbon paper electrode

## Abstract

The electrocatalytic oxidation of glycerol by metal electrocatalysts is an effective method of low-energy-input hydrogen production in membrane reactors in alkaline conditions. The aim of the present study is to examine the proof of concept for the gamma-radiolysis-assisted direct growth of monometallic gold and bimetallic gold–silver nanostructured particles. We revised the gamma radiolysis procedure to generate free-standing Au and Au-Ag nano- and micro-structured particles onto a gas diffusion electrode by the immersion of the substrate in the reaction mixture. The metal particles were synthesized by radiolysis on a flat carbon paper in the presence of capping agents. We have integrated different methods (SEM, EDX, XPS, XRD, ICP-OES, CV, and EIS) to examine in detail the as-synthesized materials and interrogate their electrocatalytic efficiency for glycerol oxidation under baseline conditions to establish a structure–performance relationship. The developed strategy can be easily extended to the synthesis by radiolysis of other types of ready-to-use metal electrocatalysts as advanced electrode materials for heterogeneous catalysis.

## 1. Introduction

The selective electro-oxidation of glycerol has been proposed as the most viable pathway for the production of value-added chemicals from waste biodiesel waste, as well as a cogeneration pathway for the production of green H_2_ using high-efficiency electrolyzers [1,2,3]. Glycerol production has been increasing since the early 2000s, which on the one hand lowers its market price, and on the other hand encourages its use as a feedstock. The selective electro-oxidation of glycerol can lead to value-added products such as glyceric acid, dihydroxyacetone, tartronic acid, glycolic acid, and formic acid with applications in many fields [4,5].

Electrocatalysts are required for reducing the activation energy of glycerol electro-oxidation. Several studies have described promising strategies for designing and constructing efficient electrocatalysts, in which the final concept combines the production of high-value chemicals with the production of hydrogen in electrolysis cells [2,4,5,6,7,8,9]. These strategies suggest considering increasing the intrinsic activity of a catalyst’s active site and increasing its number of active sites. Therefore, the design of the electrocatalyst needs a precise control of different parameters, such as surface composition, geometry, morphology, and support, as they may have an impact on glycerol’s affinity, catalytic activity, as well as oxidation selectivity. Tremendous studies have been published on electrocatalyst synthesis to improve the performance in activity and stability for glycerol oxidation. The range of materials is relatively wide, from noble metal such as Pt, Pd, and Au, and their combination with non-noble metals (Fe, Co, Ni, Ag, and Cu) in the form of monometallic, bimetallic, or tri-metallic structures [10,11,12,13,14,15,16,17,18,19]. Pt NPs were widely studied and reported as a highly active electrocatalyst for alcohol electro-oxidation. However, their scarcity and surface poisoning by carbonaceous intermediates requires searching for better alternatives. Among the different electro-catalysts, gold is one of the best alternatives to Pt due to its high stability in the electro-oxidation of glycerol under alkaline conditions and lower price. Furthermore, unlike Pt, Au is not susceptible to surface poisoning by CO adsorption, which promotes glycerol oxidation. In alkaline solution, OH groups adsorb on gold’s surface, which facilitates the adsorption of glycerol molecules onto the metal’s surface and the dissociation of the O-H and C-H bonds of glycerol [20,21]. Gold was coupled with different metals (e.g., Pd, Cu, Ag, Ni, and Co) to modify the activity and selectivity of the electrocatalyst and lower the price [11,22,23,24,25,26,27]. For instance, the deposition of copper onto gold was shown to improve the selectivity of glycerol electro-oxidation towards C3 products [23]. A beneficial electronic effect from additional Ag in glycerol electro-oxidation was shown by Garcia et al. [22] in alkaline medium. The authors demonstrated that the onset potential decreased from 1.09 V vs. RHE (reversible hydrogen electrode) for monometallic Au/C to 0.51 V vs. RHE for bimetallic AuAg/C, when coupling Au with Ag on an XC-72 Vulcan carbon substrate. Moreover, the current density of the forward scan of cyclic voltammetry (CV) in the presence of 1 mol·L^−1^ glycerol reached the maximum value of 1.60 A·mg_Au_^−1^ for AuAg, larger than other studied compositions (Au/C, Au_3_Ag/C, and Au_3_Ag/C).

Most of the methods for the preparation of gold-based electrocatalysts are multi-step procedures that involve the synthesis of gold nanoparticles in colloidal solution or powder and the post-synthesis deposition on a conductive substrate using a binding agent (e.g., Nafion) before any electrocatalytic tests [28,29,30,31]. However, a multi-step procedure complicates the preparation of large-scale electrodes. In addition, the use of a binding agent can interfere with the active sites of the catalyst and reduce its activity. Therefore, methods that allow direct growth of the gold-based catalyst on the substrate may be of interest for future electrode development for the fuel cell and electrolysis cell industries.

Several methods were explored for the one-step growth of electrocatalytic gold nanoparticles on an electrode’s surface [32,33,34]. Electrochemical deposition (e.g., galvanostatic or cyclic voltammetry) allows a quick deposition and control over the size and morphology of particles [32,35,36,37,38,39]; however, it produces the growth of non-conformal nanoparticles on porous substrates. To realize the direct in situ growth of nanoparticles over a 2D electrode substrate (e.g., carbon paper), we deploy radiolysis; that is the originality of our work. Other unconventional physical synthesis routes exist, such as microwave-assisted and sono- and mechanochemical procedures based on high temperatures and pressures, high-energy ball mills, or low-temperature ultrasonic frequencies. These procedures are effective in producing metallic nanoparticles [40,41,42] but, unlike radiolysis, they do not induce homogeneous reduction and nucleation throughout the sample volume, (solutions or heterogeneous media), leading to the homogeneous nucleation and growth of NPs at room temperature. Additionally, the high reducing power of solvated electrons enables a reduction of salts of non-noble metals (such as Fe, Ni, or Co), which are difficult to reduce by chemical methods at room temperature. Another advantage of radiolysis is that it is a powerful method to synthesize bimetallic nanoparticles of controlled size and structure (core–shell or alloys) [18,43,44,45]. To our knowledge, this is the first time that the radiolytic reduction of metal salts on a solid flat substrate has been described.

In this work, ready-to-use Au and Au-Ag nanostructured particles on a carbon fiber surface were produced by the radiolysis procedure to interrogate their electrocatalytic properties for glycerol oxidation in basic conditions [43]. The radiolysis of an aqueous solution of metal ions is known to lead to homogeneous metal nanoparticles in a single step with proper control of the nucleation process without the need for addition of chemical reducing agents [44,45,46]. High-energy radiations (such as gamma rays, X-rays, or electron beams) ionize deaerated water, generating reductive (solvated electrons and $H^{•}$ radicals) and oxidative (${\mathrm{HO}}^{•}$ radicals) species (see Equation (1)). Oxidative radicals, ${\mathrm{HO}}^{•}$, are scavenged by added secondary alcohol (see Equation (2)) or formic acid, which otherwise would cause the reverse oxidation of low-valence metal ions or formed atoms. Additionally, secondary alcohol molecules scavenge $H^{•}$ radicals while generating H_2_ and reducing alcohol radicals (see Equation (3)). Metal ions are reduced by solvated electrons (e_aq_^−^) and by reducing alcohol radicals (see Equations (4) and (5)). The reduced atoms nucleate and coalesce into stable aggregates whose size is controlled by adding stabilizing agents such as surfactants, polymers, and some ligands that adsorb to the metal’s surface.
(1)$$H_{2}O\to {\;e}_{\mathrm{aq}}^{-},{\;H}_{3}O^{+},{\;H}^{•},{\;H}_{2},{\;\mathrm{OH}}^{•},{\;H}_{2}O_{2}$$
(2)$${\left({\mathrm{CH}}_{3}\right)}_{2}\mathrm{CHOH}+{\;\mathrm{OH}}^{•}\to {\left({\mathrm{CH}}_{3}\right)}_{2}C^{•}\mathrm{OH}+{\;H}_{2}O$$
(3)$${\left({\mathrm{CH}}_{3}\right)}_{2}\mathrm{CHOH}+{\;H}^{•}\to \;{\left({\mathrm{CH}}_{3}\right)}_{2}C^{•}\mathrm{OH}+{\;H}_{2}$$
(4)$$M^{+}+{\;e}_{\mathrm{aq}}^{-}\to {\;M}^{0}$$
(5)$$M^{+}+{\left({\mathrm{CH}}_{3}\right)}_{2}C^{•}\mathrm{OH}\to {\;M}^{0}+{\left({\mathrm{CH}}_{3}\right)}_{2}\mathrm{CO}+{\;H}^{+}$$

Previous works describe the synthesis of supported metal nanoparticles by radiolysis, either from an aqueous mixture with post-synthesis deposition on conductive substrates [47,48,49,50] or from a suspension in the presence of a powder substrate [51,52,53].

A phenomenon of galvanic replacement can take place (generally the case at low dose rates) in the case of the synthesis of bimetallic Au-Ag particles that is also referred to as silver segregation [25,54,55]. Galvanic replacement occurs between reduced silver atoms and gold ions, and results in the oxidation of silver and the reduction of gold ions. Gamma radiolysis at a low dose rate of a solution containing gold (III) complexes and silver (I) ions first leads to the reduction of silver as it requires fewer transferred electrons. Then, reduced silver atoms can be replaced by gold (0) particles upon the galvanic replacement [54]. When all the gold ions are reduced, silver ions reduction takes place on the Au nanoparticles, leading to Au_core_-Ag_shell_ nanoparticles being obtained. The dose rate controls the reduction kinetics. At a high dose rate, the reduction of the metal ions is very fast and AgAu nanoalloys are obtained [54].

The aim of the present work is to examine the proof of concept for the radiolysis-assisted direct growth of gold and gold–silver nanostructured particles onto a gas diffusion electrode (GDE) to yield free-standing electrocatalysts. To prepare the direct growth of electrocatalysts on substrate, we revised the gamma radiolysis procedure to generate ready-to-use Au and Au-Ag nanostructured particles on carbon paper (CP, also referred to as GDE) by the immersion of the substrate in the reaction mixture. The metal particles were deposited on CP during radiolysis in the presence of capping agents to avoid agglomeration caused by Van der Waals forces. The electrode materials were characterized by electrochemical techniques, Scanning Electron Microscopy, and X-ray Photoelectron Spectroscopy, and tested in glycerol electro-oxidation under basic conditions to establish a structure–performance relationship. This strategy can be readily extended and further developed to other types of metal catalysts for the possible widespread deployment of radiolysis to produce ready-to-use electrocatalysts for heterogeneous catalysis.

## 2. Materials and Methods

### 2.1. Materials

Potassium tetrachloroaurate (III) hydrate (KAuCl_4_·H_2_O, assay 99%), 2-propanol (assay ≥ 99.5%), sodium citrate dihydrate (Na_3_C_6_H_5_O_7_·2H_2_O, assay ≥ 99%), poly(acrylic)acid (partial sodium salt, 60 wt.% solution in water) with average M_w_ = 2000 (GPC), silver sulfate (Ag_2_SO_4_, ass. ≥ 99%), and lead nitrate (Pb(NO_3_)_2_, assay > 99.0%) were purchased from Sigma-Aldrich. Sodium hydroxide (assay 98.70%) was obtained from Fisher Scientific (Illkirch, France) and glycerol (assay ≥ 99% extra pure) from Acros Organics (Geel, Belgium). Gas diffusion electrode (GDE) based carbon paper (further referred to as CP), with a thickness of 190 µm, was ordered from the Fuel Cell Store (Bryan, TX, USA). The used outgassing gas (argon) was of ultrapure quality and purchased from Air Liquide France (Paris, France). Ultra-pure water was provided by Milli-Q Millipore source (MQ: 18.2 MΩ cm at 20 °C).

### 2.2. Radiolytic Synthesis

Radiolytic protocol of Au NPs synthesis was based on previously published synthesis procedures in the absence and presence of supports [47,48,56]. In this work, we adapted the methodology by immersing the CP substrate in the reaction mixture during radiolysis, to obtain ready-to-use electrocatalysts. A panoramic ^60^Co gamma source located at Institute of Physical Chemistry (Orsay, France) with a maximum dose rate of 4 kGy/h was used. The CP was cut into a rectangle of size 3 × 2.5 cm and inserted into the glass vial, which was placed in front of the gamma source. A 30 mL aqueous solution already containing dissolved Au and/or Ag precursors (KAuCl_4_, Ag_2_SO_4_), 2-propanol (a scavenging secondary alcohol), and a capping agent was poured into the vial. Either sodium citrate (further referred to as Cit) or poly(acrylic)acid (further referred to as PAA) was used as a capping agent. Their concentrations were varied for the different reaction mixtures, which are described below in Table 1 for the synthesis of both monometallic and bimetallic electrocatalysts. The solution in the presence of the CP substrate was exposed to irradiation doses, defined on the basis of 4.8 kGy for the reduction of 1 mM Au (III) and 1.6 kGy for the reduction of 1 mM Ag (I). The total dose was adjusted to the concentrations of the two metals in the final mixture.

It was not possible to synthesize a monometallic Ag electrocatalyst in the presence of both PAA and Cit, or a bimetallic Au-Ag electrocatalyst in the presence of Cit (see Appendix A), due to particle agglomeration and precipitation shortly before the onset of radiolysis.

### 2.3. Characterization Techniques

#### 2.3.1. Cyclic Voltammetry

Cyclic voltammetry (CV) was carried out in a three-electrode cell in alkaline conditions. An amount of 1 M aqueous solution of NaOH was used as an electrolyte. A double-junction mercury–mercury oxide electrode (MOE) was purchased from Origalys France and utilized as a reference electrode. All CV graphs were reported with potential values converted to the reversible hydrogen electrode (RHE) scale. The conversion method is described in Appendix A. A 1 × 0.5 cm rectangular piece of the as-prepared CP was prepared as the working electrode. Note that only half of this piece (0.5 × 0.5 cm) was immersed in the solution and served as a working area, providing 0.50 cm^2^ of the geometrical surface area that received the deposition. The real surface area of the 3D microfiber structure was not considered. The second half of the CP was left for wire attachment. A glassy carbon plate of 12.4 cm^2^ was used as a counter electrode. CV measurements were conducted at a scan rate of 50 mV·s^−1^ between 0.1 and 1.6 V vs. RHE for monometallic samples. Unless otherwise specified, bimetallic samples were analyzed between 0.1 and 1.55 V vs. RHE. CV scans performed in 1 M NaOH solution without glycerol are referred to as blank CV. To evaluate the glycerol oxidation activity, CV scans were performed in the aqueous solution of 1 M NaOH + 0.1 M glycerol. The blank and glycerol voltammograms were corrected for iR drop. The iR-free graphs were plotted by replacing the *E*_applied_ with *E*_real_ based on the expression *E*_real_ = *E*_applied_ − *R*_Ω_ × *I*. The resistance *R*_Ω_ is defined by the intersection of the Nyquist curve with the *X*-axis at high frequencies in electrochemical impedance spectroscopy (EIS) analysis. The calculation of the electrochemically active surface area is provided in the Appendix A.

#### 2.3.2. Electrochemical Impedance Spectroscopy (EIS)

Electrochemical impedance is identified by measuring current upon applying an excitation potential that generates an alternating current at the same frequency, ω, as the potential. The data points of impedance (Z(ω)) in EIS are produced at different frequencies as real (Z′) and imaginary (Z″) components of impedance. Setting the real component on an *X*-axis and the imaginary component on a *Y*-axis builds up a semi-circle shape on a Nyquist plot.

The charge-transfer resistance (R_ct_) is an important parameter that gives an understanding of the ability of the electron transfer electrode reaction to conduct a large current density with a smaller driving force (overpotential). The R_ct_ is defined by the size of the semicircle in the Nyquist plots [57]. The measurements were carried out at different electrode potentials in the range of frequencies between 100 kHz and 25 mHz. The amplitude was set to 10 mV.

#### 2.3.3. Underpotential Deposition of Lead (Pb UPD)

Pb UPD analysis was performed to identify the crystallographic orientation of grown monometallic Au and bimetallic Au-Ag particles. The deposition of Pb on the radiolysis-modified carbon paper electrode surface (1 × 1 cm square) was realized by cyclic voltammetry scans in 1 mM Pb(NO_3_)_2_ and 1 M NaOH solution. The potential was cycled from 800 to 250 mV vs. RHE in the three-electrode cell in the presence of the MOE reference electrode and the glassy carbon plate (12.4 cm^2^) as a counter electrode.

#### 2.3.4. Material Characterization

Scanning Electron Microscopy (SEM) micrographs were obtained on a ZEISS Hitachi S-4800 microscope. To enhance the high-resolution (HR) SEM imaging capability, a thin layer (1–5 nm) of carbon or platinum was coated on the samples. Energy dispersive X-ray (EDX) analysis was carried out with EVOHD 15 microscope.

X-ray Photoelectron Spectroscopy (XPS) analysis was conducted on a PHI 5000 Versa probe II apparatus from ULVAC-PHI Inc. A monochromatized Al Kα source (1486.6 eV) was used with a spot size of 20 μm. A charge neutralization system was used to limit a charge effect. The remaining charge effect was aligned, fixing the C–C bond contribution of a C 1s peak at 284.8 eV. Survey spectra were recorded with a 187 eV pass energy while high-resolution spectra were recorded with a 23 eV pass energy. All the peaks were fitted with Casa XPS software (version 2.3.24PR1.0) using a Shirley background. Quantification from deconvoluted spectra was carried out using the transmission function of the apparatus and angular distribution correction for an angle of 45°. Sensitivity factors were extracted from their integration of cross-section and escape depth correction.

Inductively coupled plasma optical emission spectrometry (ICP-OES) analysis was performed on 1 × 1 cm of carbon paper after deposition of Au or Au-Ag NPs. Prior to the measurement, samples were submerged in a mixture of concentrated HNO_3_ (70%) and HCl (37%) (*v*/*v*. 1:1). The complete solubilization of the metals was achieved by heating the whole mixture of the acids and electrocatalyst with microwaves in a sealed reactor for 30 min. A spectrometer Optima 2000 DV from Perkin Elmer was used to conduct a quantitative determination of metal weight fractions.

X-ray Diffraction (XRD) data were collected at IEM with an X’pert Pro 2-circle diffractometer equipped with an X’celerator detector. The diffractogram was recorded in Bragg–Brentano configuration with Ni-filtered Cu K-alpha radiation.

## 3. Results and Discussion

### 3.1. Characterization of the Gold-Based Electrocatalysts

#### 3.1.1. SEM and EDX Analysis

The successful synthesis of metallic gold formed by direct growth on carbon paper by immersing the support in the reaction mixture during gamma radiolysis was demonstrated by SEM observations (Figure 1).

Monometallic Au particles were deposited on the CP surface from a 1 mM KAuCl_4_ solution stabilized by different concentrations of Cit (1.3 mM or 40 mM in the first-row images) or PAA (0.5 M or 1.0 M in the second-row images). The resulting materials are referred to as Au_1.3mM-Cit, Au_40mM-Cit, Au_0.5M-PAA, or Au_1M-PAA, respectively. At lower resolutions, SEM images display successful growth of metal particles on the outer as well as inner fiber layers (a few fibers in depth). The species produced upon radiolysis, solvated electrons and alcohol radicals, reduce the metal ions or complexes (which have diffused in the CP) very homogeneously in the substrate [32]. Therefore, the radiolysis method results in the formation of metal particles on the upper and inner layers of the fibers, in contrast to the electrochemical electrodeposition method that leads to the growth of metal particles only on the outer fibers [46].

For Au_1.3mM-Cit, the higher magnification image (scale bar of 1.2 µm) shows the homogeneous distribution of the nanoparticles. Two different particle morphologies are observed: flower-like particles formed by several 2D platelets with an overall size of 1–1.5 µm, and nanoparticles characterized by individual platelets of about 40 nm. However, the addition of a higher amount of Cit up to 40 mM significantly lowers the concentration of Au NPs on the surface (images in the top right corner). It has been shown that sodium citrate is able not only to stabilize Au NPs but also to induce the reduction of Au (III) ions in solution [58]. This reduction is facilitated when a solid substrate (e.g., CP) is present in the reaction mixture. Therefore, a reduction of AuCl_4_^−^ is probably initiated by the citrate prior to gamma irradiation and inducing the adsorption of Au colloids on the CP surface.

Radiolysis of the KAuCl_4_ solution in the presence of PAA generated rather different Au particle morphologies. The PAA concentration between 0.5 M and 1 M led to the formation of two groups of particles, one ranging from 270 to 310 nm and the second in the 100 nm range (see second row in Figure 1). The density and size distributions of the particles remain non-homogeneous.

An electrocatalyst consisting of a silver deposit alone was not reported in this work due to the infeasibility of its synthesis under the studied conditions (more details in the Materials and Methods section) and its low activity towards glycerol oxidation known from the literature [25,30].

Au-Ag bimetallic particles were grown in situ by radiolysis of the reaction mixtures containing the dissolved precursors of [AuCl_4_]^−^ and [Ag]^+^ in a molar ratio of 75:25 or 50:50 (referred as to Au_75_/Ag_25_ and Au_50_/Ag_50_, respectively), while the total concentration of metal ions was maintained constant at 2 mM. They were synthesized in the presence of PAA with the concentrations 0.5 or 3.2 M PAA. A significant agglomeration of the particles is observed with a very wide size distribution ranging from a few tens of nanometers to several micrometers (see third- and fourth-row images in Figure 1). As the reaction mixture is prepared using dissolved metallic precursor salts, the counter ions may establish an electrolytic environment that favors the dehydration of the PAA layer, leading to its chain shrinkage and particle aggregation [59]. An increase in particle density was observed with increasing Ag concentration in the initial reaction mixture (from 75:25 to 50:50) and increasing PAA concentration from 0.5 M to 3.2 M. Sodium citrate was also tested in the fabrication of Au-Ag particles; however, rapid precipitation when Cit was added to the reaction mixture prior to radiolysis did not allow for the radiolytic synthesis of Au-Ag particles on the CP surface.

The EDX spectra and calculated atomic fractions support the relative increase in the Ag fraction in the final materials with the increase in PAA concentration (see Appendix A and Appendix A). When PAA increases from 0.5 M to 3.2 M, the Au:Ag ratio varies from 20:1 to 15:1 for the Au_75_/Ag_25_ sample, and from 10:1 to 2:1 for the Au_50_/Ag_50_ sample. These variations confirm that a higher amount of PAA leads to a higher fraction of Ag adsorbed on the CP surface compared to Au. The galvanic replacement phenomenon, leading the reduced silver (Ag^0^) to react with the Au (III) species to form Au^0^ and Ag (I), could be a reason for the high Au/Ag ratio. Such galvanic replacement leads to the eventual reduction of the gold precursor rather than silver in the initial stages of radiolysis [54].

In this work, the Au particles reached a much larger size, between 100 nm and 1300 nm, depending on the nature of the capping agent, with a very wide size distribution as compared with 2–20 nm Au synthesized by gamma radiolysis reported in the literature [43,46,52,60]. In most previous studies, Au NPs were deposited in situ or by subsequent wet impregnation on powder substrates using templates or binders. We have provided the first proof of concept to directly grow gold–silver-based nanostructured materials by radiolysis onto gas diffusion electrodes (GDEs) as free-standing electrocatalysts. Further optimizations, including the impact of irradiation dose, dose rate, and pH [31,32], should be considered in future work to decrease the particle size and reduce the size distribution.

#### 3.1.2. Electrochemical Analysis

To determine the electrochemical characteristics of the as-fabricated materials, we next utilized the CV in 0.1 M NaOH solution (further referred to as blank CV). For the monometallic samples, the blank CV profiles shown in Figure 2A display a capacitive current range between 0.1 and 0.6 V vs. RHE and a broad anodic peak around 1 V vs. RHE, which is attributed to the specific adsorption of OH^−^ species (i.e., formation of surface Au(OH)* species) [61,62]. The following increase in the anodic current around 1.3 V vs. RHE samples occurs due to oxidation of Au(OH)* to AuO_x_ oxide [61,63]. The potential sweep was reversed at 1.6 V vs. RHE prior to further oxidation of the water. An asymmetric cathodic peak was observed at 1.07 V vs. RHE, assigned to the reduction of gold oxide back to metallic gold. For the sample Au_1.3mM-Cit firstly synthesized (black dash line), the increase in negative current at low potential is assigned to the reduction of O_2_ at the surface, which originates from a poor removal of the dissolved gas by bubbling N_2_ gas in the electrolyte that was further improved for the following samples. For the sample Au_40mM-Cit (red dash line), a greater Cit induces a current density that is about four times lower, and it is consistent with the reduced deposition density observed in the SEM micrographs. For samples prepared with PAA (blue and green dash lines), the voltammograms show broader oxidation peaks which are probably due to the altered access and diffusion of solutes at the metal surface [64,65]. The current density peaks increase with increasing PAA concentration, consistent with the deposition density observed in the SEM images. The reduction peak shows a shoulder on the more negative potential side, suggesting the possible formation of different Au crystal lattices, as evidenced by the different Au morphologies (small particles and larger agglomerates).

In the case of bimetallic CP-Au-Ag samples, a capacitive current is observed between 0.1 and 0.7 V vs. RHE on CV profiles (Figure 2B). A slight increase in current between 0.90 and 1.15 V vs. RHE is assigned to the specific adsorption of OH^−^ species on the Au-Ag surface [63,66]. The anodic current increases from 1.20 V vs. RHE and diminishes to about 1.52 V vs. RHE. The current density peaks decrease in the following order of samples: Au_50_/Ag_50_-3.2M-PAA > Au_75_/Ag_25_-0.5M-PAA > Au_75_/Ag_25_-3.2M-PAA > Au_50_/Ag_50_-0.5M-PAA, contrary to the deposition density observed in SEM images. This may be attributed to an inhomogeneous distribution of the particles deposited on the outer surface and inner fibers and to the instability of the PAA-capped particles. Nevertheless, the positions of the peaks provide useful information about the electrode surface. The samples Au_75_/Ag_75_-3.2M-PAA (red solid line) and Au_50_/Ag_50_-3.2M-PAA (green solid line) exhibit an oxidation peak at 1.37 V vs. RHE with a shoulder at 1.28 V vs. RHE. This peak shift to a more positive potential, as compared to the monometallic Au samples, can be linked to the presence of silver and PAA on the surface. The shoulder can be attributed to the late oxidation of Au → Au(OH)*, while the peak can be assigned to the further oxidation of gold and silver to oxides, AuO_x_ and Ag → Ag_2_O. The anodic peak of Au_75_/Ag_25_-0.5M-PAA (black solid line) is even more shifted towards the higher potential of 1.47 V vs. RHE, with a shoulder at 1.34 V vs. RHE. The shoulder can be ascribed to the oxidation of Au and Ag to their oxides, while the main peak is a sign of the oxidation of residual silver [61]. The presence of a residual fraction of Ag is supported by the highest Au:Ag ratio in Au_75_/Ag_25_-0.5M-PAA defined by EDX (i.e., lower Ag content). A very low and broad anodic peak observed for the Au_50_/Ag_50_-0.5M-PAA (blue solid line) points to a low loading of Au and Ag NPs or serious coagulation, leading to a decrease in the active surface area.

Reversing the potential scan at a higher potential of 1.6 V vs. RHE for the Au_50_/Ag_50_-3.2M-PAA sample induced further oxidation of surface species. As no such oxidation was observed for the monometallic radiolytic samples, the origin of this second peak can be ascribed to the oxidation of silver (I) oxide to silver (II) oxide. This silver (II) oxide species are then reduced back to silver (I) oxide at 1.34 V vs. RHE (see green line). When the scan was reversed at 1.55 V vs. RHE for the other bimetallic samples, an analogous second electron exchange was not observed. The following narrow cathodic peak appears at 1.12 V vs. RHE for Au_75_/Ag_25_-3.2M-PAA (red line), which is assigned to the reduction of Ag_2_O to Ag^0^. The shoulders of these peaks are at 1.02 V vs. RHE resulting from the reduction of AuO_x_ to Au. The high current density generated by the narrow part of the peak supports the segregation of Ag particles to the outer shell. In contrast, the Au_75_/Ag_25_-0.5M-PAA (black line) and Au_50_/Ag_50_-0.5M-PAA (blue line) samples display a little bump on the peak shoulder, pointing to a smaller Ag surface area in agreement with EDX analysis. Their peak maxima are at 1.02 V vs. RHE and support a larger Au surface area. The other sample, Au_50_/Ag_50_-3.2M-PAA (green line), shows a broad peak at 1.06 V vs. RHE that is attributed to the reduction of AuO_x_ and Ag_2_O, pointing to the presence of an alloyed structure.

Monometallic Au particles were obtained using PAA and Cit as capping agents, whereas bimetallic Au-Ag particles were obtained only with PAA (rapid precipitation of Au-Ag in the presence of Cit). Therefore, the difference in the impact of PAA or citrate on the final electrochemical properties can only be evaluated for Au particles. The higher concentration of PAA promoted a higher deposition density of the particles, which correlates with the greater reduction peak. However, a higher concentration of Cit led to lower Au deposition density on the CP and a low reduction peak, which may be related to a higher concentration of remaining undeposited Au particles. The sample with a lower concentration of Cit, Au_1.3mM-Cit, led to higher deposition density and a higher reduction peak. A similar influence of PAA was observed for bimetallic samples. The higher concentration of PAA led to higher reduction peaks of bimetallic samples, which is also consistent with the greater deposition densities in Au_75_/Ag_25_-3.2M-PAA and Au_50_/Ag_50_-3.2M-PAA.

#### 3.1.3. X-ray Photoelectron Spectroscopy

We used XPS to probe the surface state and composition of the bimetallic Au_75_/Ag_25_-0.5M-PAA, Au_50_/Ag_50_-0.5M-PAA, and Au_50_/Ag_50_-3.2M-PAA samples more closely. For all three samples, survey XPS analysis (Appendix A) confirms the presence of gold, silver, carbon, and oxygen atoms (more detailed information on the XPS peaks from the survey analysis can be found in the Appendix A) [67].

For the Au_75_/Ag_25_-0.5M-PAA sample, the high-resolution XPS spectra of typical Au 4f and Ag 3d orbital ranges show the formation of the metallic Au and Ag states with the Au 4f_7/2_ peak at 84.4 eV and Ag 3d_5/2_ peak at 368.2 eV (see Figure 3a). Furthermore, the observation of O 1s peaks at 533.4 eV in the high-resolution XPS spectra (see Appendix A) corroborates the absence of metal oxides that generate a characteristic signal below 531 eV [67]. The data of deconvoluted XPS spectra are reported in Appendix A. The observation of the peak positions is consistent with XPS reports for supported Au^0^ and alloyed Ag^0^ structures [68,69,70,71], while they disagree with XPS reports of their oxidized forms [67,72,73]. However, the large FWHM of the Ag 3d signals suggests the presence of another component at a higher binding energy. We hypothesized the emission of Au 4d_5/2_ and 4d_3/2_ plasmons at 370.6 and 376.8 eV, respectively. The observation of this plasmon effect under X-ray beam suggests the presence of gold particles of nanometric size on the surface. The deconvolution of the peaks resulted in a Au:Ag atomic ratio of 2.2:1. An important difference between the Au:Ag atomic ratios defined by XPS (2.2:1) and EDX (20:1) in Au_75_/Ag_25_-0.5M-PAA suggests the segregation of silver atoms on the upper surface of Au-Ag particles (see comparison of metal fractions by XPS and EDX estimates in Table 2). This is related to the different depth of analysis of the two instruments. The XPS analysis provides the compositional information only for the top thin layer (4–10 nm) and EDX characterizes the bulk by penetrating much deeper (>1 µm). Therefore, the different Au:Ag ratios obtained by the two instruments confirm that the Ag content increases from bulk to the top surface with respect to the Au content. Therefore, it suggests that Ag is more concentrated at the outer surface of Au-Ag particles, pointing to a possible segregation of silver. This is consistent with the reported Ag segregation in the radiolytic synthesis of Au-Ag bimetallic nanoparticles [54].

For the Au_50_/Ag_50_-0.5M-PAA sample, with equal molar fractions of [AuCl_4_^−^] and [Ag^+^] in the initial reaction mixture, the high-resolution XPS spectra (see Figure 3b) displays narrow-scan spectra of Au 4f and Ag 3d at the same positions, confirming the metallic state of both metals.

The resulting deconvolution of these spectra revealed a Au:Ag atomic ratio of 5.4:1 (Appendix A), which is twofold lower than the value defined by EDX (10:1). This suggests that gold is enriched more in the core of the Au-Ag particles rather than in the top layer of the particles (see Figure 4). It also points to the difference in the reduction rates of Au (III) in the initial and final stages of the radiolysis. In the initial stages of the radiolysis, reduced silver promotes the reduction of gold and oxidizes back to silver (I), contrary to the final stages. Moreover, the Ag 3d region in XPS also shows the presence of plasmons from the Au 4d_5/2_ and 4d_3/2_ components similarly to Au_75_/Ag_25_-0.5M-PAA, which supports the presence of metallic Au on the top surface.

Finally, for the Au_50_/Ag_50_-3.2M-PAA sample, the high-resolution XPS measurement also exhibits Au 4f and Ag 3d at the analogous positions consistent with the metallic state (Figure 3c). Herein, the ratio of atomic fractions Au:Ag decreased to 1.1:1 (Appendix A). This is consistent with increased concentrations of Ag (I) (from 0.5 mM to 1 mM) and PAA (from 0.5 M to 3.2 M) in the reaction mixture. The analogous decrease in the Au:Ag ratio was also observed by EDX analysis (1.6:1). Therefore, it suggests that the Ag content is slightly more concentrated near the upper surface of the particles rather than in the core. Such segregation of Ag to the top surface and low Au:Ag ratio (1.1:1) could be a reason for the absence of the Au 4d plasmon signal.

High-resolution XPS analysis of two equimolar (Au_50_/Ag_50_) samples confirms the reduction of Au and Ag to the metallic states and corroborates the segregation of Ag atoms. It can be proposed that promotion of Au (III) reduction by Ag segregation is affected by the initial ratio of Au and Ag precursors. When the initial Au:Ag ratio is low, the reduction of gold occurs at a higher rate than the reduction of silver due to two processes: radiolytic effect (reduction by solvated electrons and reducing radicals) and galvanic replacement with the silver atoms. This results in a silver-rich shell on the bimetallic particles. When the initial Au:Ag ratio is high (e.g., Au_75_/Ag_25_), two mentioned paths for gold’s reduction are also valid, but the galvanic replacement with the lower concentration of silver may require a higher number of reduction/re-oxidation cycles and contribute less to the overall reduction of gold. A high difference in the Au:Ag ratios (2.2 vs. 20) in the outer shell and in the whole particle of Au_75_/Ag_25__0.5M-PAA observed by XPS and EDX, respectively, suggests that there were a high number of redox cycles of silver followed by its segregation.

A high-resolution XPS study of the Au 4f spin-orbitals in all the samples revealed 4f_7/2_ and 4f_5/2_ components (84.4 and 88.2 eV, respectively) matching well with metallic Au [70,74] and locating at a lower binding energy than Au(I) or Au (III) (4f_7/2_ > 85 eV) [67]. Analogously, the positions of silver 3d_5/2_ and 3d_3/2_ peaks supported metallic states and excluded the formation of silver oxide or halide. The deconvolution of the XPS data and the comparison with EDX results have pointed to the higher concentration of gold content in the core and lower concentration of gold in the upper surface of the particles, which supports the phenomenon of silver segregation and the formation of core–shell Ag@Au nanoparticles.

#### 3.1.4. X-ray Diffraction

X-ray Diffraction (XRD) analysis reveals metallic structures on the surface of the monometallic and bimetallic samples (see Appendix A). The samples Au_1M-PAA, Au_75_/Ag_25_-0.5M-PAA, and Au_50_/Ag_50_-0.5M-PAA show (111), (200), (220), and (331) crystal lattice patterns of the *fcc* crystal phase at 38.4°, 44.7°, 64.8°, and 77.7°, respectively. These 2θ values are in agreement with the XRD data of Au and Ag particles reported by our group and other research groups [33,34]. These peaks are observed with a shift from the reference XRD patterns of Au in the *fcc* phase (JCPDS 00-004-0784, gray bars in Appendix A) expected at 38.2°, 44.4°, 64.6°, and 77.5°. The *fcc* structure of Ag exhibits XRD patterns very close to those of the *fcc* Au (JCPDS 00-004-0783, green bars in Appendix A). The 2θ shift between the reference and actual peaks is about 0.2°, which might be related to the highly contrasting nature and thickness of materials under an X-ray beam. It is also termed as a problem of X-ray micro absorption. Reference patterns of carbon (see orange bars corresponding to JCPDS, 00-026-1080) helped to assign XRD peaks at 43.3° (with a shoulder at 42.7°), 47.7°, 54.8°, and 65.8° to the following carbon lattices: (101), (100), (103), (008), and (107), respectively. The other peaks that were observed at 36.2°, 39.6°, 48.7°, and 57.6° remained unassigned. The presence of AgCl can be excluded as the XRD patterns would show more shifted peaks (ca. 32°, 46°, 55°, and 57.5°) [75,76]. However, the sample Au_50_/Ag_50_-3.2M-PAA (brown curve), obtained with a higher PAA agent, displays less intense peaks of initially defined crystal lattices, such as (111), (200), and (220). This might originate from the use of a six-fold higher concentration of the PAA agent in the reaction mixture, which could affect the nucleation step of Au and Au-Ag particles. However, the unassigned peaks are still observed with almost the same intensity as the first two samples.

By considering the intensities and areas of the peaks, the Pb_UPD_ characteristics clearly reveal the formation of a gold metal surface upon gamma irradiation for the samples Au_1.3mM-Cit and Au_1M-PAA (in Appendix A and Appendix A, respectively). The peaks at 0.38 and 0.43 V vs. RHE are assigned to the adsorption and desorption of the Pb monolayer onto the (111) facets, respectively, and the large anodic peak at 0.60 V vs. RHE and its cathodic side at 0.52 V vs. RHE confirm a deposit on the (110) lattice. These assignments are in agreement with previous works [77,78]. The Pb_UPD_ characteristics for the bimetallic Au_75_/Ag_25_-0.5M-PAA and Au_50_/Ag_50_-0.5M-PAA (Appendix A and Appendix A, respectively) also reveal the presence of a gold metal surface with the formation of (111) and (110) crystal facets [66,79]. However, Au_75_/Ag_25_-0.5M-PAA shows a larger crystal lattice contribution (110), which could be related to a higher active surface area, whereas Au_50_/Ag_50_-0.5M-PAA shows a smaller crystal lattice contribution (110), which could be related to a smaller active surface area, in agreement with the blank CVs.

The direct growth of monometallic Au and bimetallic Au-Ag particles directly on the 3D carbon paper was achieved by in situ gamma radiation. The Au and Au-Ag particles formed a large average size (between 0.1 and 1.5 µm) and agglomeration as observed in SEM images. Moreover, the difference between Au:Ag ratios provided by XPS and EDX measurements in some bimetallic samples pointed to the segregation of Ag atoms to the upper surface.

### 3.2. Electrocatalytic Performance towards Glycerol Oxidation

#### 3.2.1. Cyclic Voltammetry Measurements in the Presence of Glycerol

We next investigated the efficiency of the samples for glycerol electro-oxidation by CV measurements in 1.0 M NaOH + 0.1 M glycerol. The CV profiles of the positive scan only are reported in Figure 5 (the negative CV scans were omitted to simplify the comparison of voltammograms to positive scans). The current density starts to increase between 0.4 and 0.7 V vs. RHE due to the oxidation of glycerol and decreases from 1.35 V vs. RHE due to the formation of an inactive oxide layer.

For the monometallic samples, the citrate-stabilized samples, Au_1.3mM-Cit and Au_40mM-Cit, generate a current density of 75.6 and 48.8 mA·cm^−2^ (at 1.28 V vs. RHE), and exhibit an onset potential at 0.65 and 0.80 V vs. RHE, respectively. The lower concentration of citrate promotes a larger electrochemical surface area and a lower onset potential. However, for the PAA-stabilized samples, higher PAA concentrations from 0.5 to 1 M promote higher current densities and lower onset potential: the sample Au_0.5M-PAA generates a current density of 63.7 mA·cm^−2^ at 1.28 V vs. RHE, whereas the sample Au_1M-PAA exhibits a current density of 89.8 mA·cm^−2^ at the lower potential 1.19 V vs. RHE (E_onset_ = 0.60 V vs. RHE). Such a favorable enhancement of the activity could originate from higher loading of Au with addition of higher PAA concentrations.

The Au-Ag bimetallic particles grown on CP by gamma radiation generated a high current density. As shown in Figure 6, the Au_75_/Ag_25_-0.5M-PAA and Au_50_/Ag_50_-0.5M-PAA materials achieve the highest current densities of 110 and 103 mA·cm^−2^, respectively, at a relatively high peak potential of 1.32 V vs. RHE. For the materials prepared with a higher concentration of PAA, Au_75_/Ag_25_-3.2M-PAA and Au_50_/Ag_50_-3.2M-PAA, lower current density peaks are observed (ca. 83 mA·cm^−2^) at a lower potential ca. 1.21 V vs. RHE. The onset potential is very similar (ca. 0.50–0.55 V vs. RHE) for all bimetallic gamma samples, and is lower than for the monometallic CP-Au samples.

#### 3.2.2. Electrochemical Impedance Spectroscopy (EIS)

Electrochemical impedance spectroscopy measurements were conducted to evaluate the charge transfer resistance (R_ct_) during the electro-oxidation of glycerol. Electrochemical impedance was measured at different applied potential values such as open-circuit potential (OCP), 0.84, 0.89, 0.94, and 0.99 V vs. RHE. The fit of the Nyquist plots by the equivalent electrochemical circuit R_Ω_ + Q_CPE_//R_ct_ enables the extraction of the R_ct_, the charge transfer resistance, with R_Ω_ the ohmic resistance or the uncompensated resistance, and with Q_CPE_ the constant phase element to model the “imperfect” capacitance of the double layer. Since the OCP value was different for each sample, the Nyquist plots were compared at the applied potential of 0.84 V vs. RHE (the lowest applied potential after OCP) in Appendix A. The R_ct_ values for the different electrodes derived from the fitting and OCP values are reported in Table 3. These R_ct_ values determined at different potentials are used to plot the EDX slope. From the first sight, one can notice that bimetallic samples provide lower charge transfer resistance as compared to any of the monometallic electrodes in this work. The monometallic Au_1.3mM-Cit is characterized by a lower R_ct_ than that of the other monometallic samples, the highest R_ct_ value being for the sample Au_40mM-Cit. As observed on CV profiles and SEM images, increasing Cit concentration leads to poor loading of Au onto the CP electrode, which could be an origin of the high R_ct_, whereas higher concentrations of PAA (from 0.5 to 1 M in monometallic samples) promote greater Au loading and thus lower R_ct_ values.

For bimetallic samples, the higher concentration of PAA also led to lower R_ct_, which is consistent with the electrochemical activities observed by CV. Apart from increasing the overall metal loading, this can also originate from higher Ag fractions as is observed for Au_50_/Ag_50_-3.2M-PAA. However, one can notice low loading in the SEM images of Au_50_/Ag_50_-0.5M-PAA that could lead to a higher R_ct_ than that of other bimetallic samples.

Tafel slopes were obtained by plotting the applied potentials as a function of the inverse values of corresponding R_ct_ (i.e., E vs. R_ct_^−1^) and log fitting the data points (Appendix A). The Tafel slopes are commonly used to semi-quantify the reaction rates occurring at the electrode–electrolyte interface [80]. Qualitatively, the Tafel slope value (given in mV·dec^−1^) is an indicator of the reaction mechanism [80,81,82] that shows how much potential is required to increase the rate of the electron exchange rate between the electrode surface and a solute molecule by one decade. Tafel slope values, collected in Table 3, show that the efficiency of the catalysts increases in the following order: Au_0.5M-PAA < Au_1M-PAA < Au_75_/Ag_25_-3.2M-PAA < Au_50_/Ag_50_-3.2M-PAA < Au_40mM-Cit < Au_1.3mM-Cit < Au_50_/Ag_50_-0.5M-PAA < Au_75_/Ag_25_-0.5M-PAA.

The trend of Tafel slopes partially correlates with the trend of peak current density values observed in Figure 5. The lowest Tafel slopes are obtained for the two bimetallic samples, Au_50_/Ag_50_-0.5M-PAA and Au_75_/Ag_25_-0.5M-PAA, which exhibited the highest peak current densities towards glycerol oxidation.

#### 3.2.3. Inductively Coupled Plasma—Optical Emission Spectroscopy

Inductively coupled plasma—optical emission spectroscopy (ICP-OES) measurements were performed to identify the Au and Ag loading in the monometallic and bimetallic electrodes, reported in Table 4 along with the calculated specific peak current density in A·mg^−1^. The results reveal that in monometallic Au electrocatalysts, PAA promotes a twofold lower loading than when sodium citrate is used. Nevertheless, the mass efficiency (A·mg^−1^) by PAA-stabilized samples remains higher compared to citrate-stabilized electrodes (8 A·mg^−1^ for Au_1M-PAA vs. 3.3 A·mg^−1^ for Au_1.3mM-Cit).

Bimetallic samples contain much higher amounts of Au (between 40.9 and 102.6 µg/cm^2^) that could originate from higher initial concentration of precursors and the presence of Ag. The Au and Ag loadings provided by ICP-OES measurements are used to determine the Au:Ag atomic ratios in bimetallic samples (see Table 4). The results are consistent with the EDX analysis, suggesting that the metal loading, especially for Ag, increases between 2 and 12 times with increasing PAA concentrations between 0.5 and 3.2 M. Furthermore, increasing the Ag content leads to a 1.5-fold lower peak current density for the initial ratio [Au]:[Ag] = 75:25, and 4-fold lower for the initial ratio [Au]:[Ag] = 50:50. This behavior may be related to the enhanced agglomeration observed in the SEM images or to the segregation of the less active Ag on the outer surface of the particles.

Nevertheless, these bimetallic samples prepared by gamma irradiation outperform some of the works in the literature (0.6–1 A·mg^−1^) [17,83] in terms of the calculated specific current density per mg of loaded metal. The recently published work of Boukil et al. [25] in our group reported the enhanced electrochemical performance of AuAg alloyed nanocages up to 18.2 A·mg^−1^, which was at that time the highest current density in the electro-oxidation of glycerol. Such a high performance in that work could originate from the high surface density of grown AuAg nanoparticles and high surface area of the electrocatalyst. These qualities of the electrode should be targeted during the mentioned optimization of the conditions for radiolysis reduction to achieve the best possible performance in glycerol electro-oxidation.

#### 3.2.4. Electrocatalytic Performance in an H-Type Cell

Preliminary studies of chronoamperometric performance and the selectivity of the electrocatalysts for glycerol electro-oxidation were carried out in an H-type cell shown in Appendix A. The half of the cell with the working electrode was filled with 1 M NaOH solution containing 0.1 M of glycerol, and the other half was filled with 1 M NaOH solution. The potential of 1.14 V vs. RHE was applied for all runs. The potential value was chosen to be slightly lower than the peak potential based on the CV profiles (Figure 5) to analyze the products at a high activity.

The chronoamperometry plots, reported in Figure 6, show that the bimetallic Au-Ag electrocatalysts (solid lines) maintain a higher current density over a longer period than the monometallic electrocatalysts. These features should result in a higher conversion. Upon applying potential, the monometallic CP-Au shows low current densities (below 16 mA·cm^−2^), decreasing to 0.5 mA·cm^−2^ within 15 min. In contrast, the bimetallic Au-Ag particles generate high current densities and display greater stability during chronoamperometry measurement. Only Au_50_/Ag_50_-0.5M-PAA among the bimetallic samples shows a sharp loss of activity (ca. 85% loss in 30 min), which is probably related to insufficient loading. The short catalytic activity of some samples originates from surface poisoning by intermediate oxidation products and the formation of gold oxide [84,85]. Bimetallic Au-Ag electrocatalysts perform better than monometallic Au in the long term, most probably thanks to the electronic and structural features of the Au-Ag alloy on the surface. It is reported that Ag empties the Au 5d band in the Au-Ag electronic shell, and consequently facilitates the desorption of intermediates from the surface [30]. In turn, it refreshes the Au surface, minimizes the surface poisoning, and prolongs the conversion rate.

The small drop in current densities for all samples at the 15th minute is due to the removal of an aliquot of the reaction mixture. Aliquots were analyzed by high-performance liquid chromatography (HPLC) after 15 and 30 min of chronoamperometry measurement by comparing with five possible reference compounds (shown in Figure 1). HPLC detects two major products: formic (48.9–63.7%) and glycolic (35.3–45.6%) acids. The identified products of the oxidation with their concentrations and calculated selectivity are reported for the monometallic and bimetallic catalysts, respectively, in Appendix A. We note that the high selectivity demonstrates that the present electrocatalysts can be implemented in an electrolyzer for formic acid production given its high potential as a fuel or chemical (for example the formic acid fuel cell). A bar plot in Figure 7 compares the selectivity of five electrodes (Au_1.3mM-Cit, Au_1M-PAA, Au_75_/Ag_25_-0.5M-PAA, Au_75_/Ag_25_-3.2M-PAA, and Au_50_/Ag_50_-0.5M-PAA).

The formation of these C2 and C1 products suggests a significant C-C bond cleavage at these conditions of electro-oxidation. Selectivity towards C3 products is found at 2.2% by both monometallic samples, whereas bimetallic samples keep their C3 selectivity below 1.3%. One of the monometallic samples, Au_1.3mM-Cit, shows the highest selectivity towards the formation of oxalic acid at about 5% among analyzed electrodes.

The presence of silver traces on the surface of bimetallic samples could be a reason for the low yield of C3 products as electrochemical measurements (chronoamperometry, cyclic voltammetry, and electrochemical impedance) showed increased current density and reduced charge transfer resistance. In addition, silver can affect the affinity of glycerol and oxidation intermediates with the surface. Nevertheless, considering the observed relatively high selectivity towards formic acid as identified by HPLC, bimetallic samples are of high interest to be further investigated at lower potentials. At lower potentials, the rate of electro-oxidation should decrease to give more time for the desorption of C3 products before cleaving a C-C bond, as described in [86,87].

The results of this work were compared to other literature precedents in Table 5. Although it is difficult to make direct comparison with other works (due to the different conditions of synthesis, metal loading, and measurement), the values of onset potential (E_onset_), peak current density (j_p_), and selectivity can give a general understanding of the relative performance. Our tests were performed at a much lower glycerol concentration of 0.1 M. As the glycerol concentration directly influences the measured current, the normalization of jp values as a function of glycerol concentration would highlight the very good performance of the bimetallic electrocatalysts prepared by radiolysis. In addition, similar to that observed in [22,88,89], bimetallic particles contribute to decreasing the onset potential of glycerol oxidation to lower values. The selectivity of monometallic electrodes towards the formation of the primary alcohol glyceric acid was dominant for Au supported on different supports [86,90] (at +1.9 V and +1.6 V vs. RHE, respectively), and could be explained by weaker interaction between the nanoparticles and glyceric acid, allowing its desorption and hindering its further decomposition into two and one carbon products. For the direct radiolysis-assisted growth of monometallic and bimetallic gold–silver nanostructured particles, performed at a lower potential of +1.14 V vs. RHE, the interaction between the particles and the support could contribute to a different selectivity of glycerol electro-oxidation towards a significant cleavage of the C-C bond and the main production of two- and one-carbon products.

## 4. Conclusions

In conclusion, we have provided the first proof of concept that in situ radiolysis can lead to electroactive Au and Au-Ag catalysts on upper and inner layers of CP fibers for glycerol oxidation. Our preliminary findings have shown that the Au and Au-Ag particles formed large average sizes, from tens to hundreds of nanometers, and agglomeration as observed in SEM images. Sodium citrate and poly(acrylic) acid were utilized to control the size and morphology of particles.

Citrate was shown to act as a capping as well as a reducing agent. The use of a relatively low concentration of 1.3 mM afforded the deposition of spherical and flower-shaped Au nano- and micro-particles on the CP surface. However, the increase in citrate concentration to 40 mM led to lower loading and deposition density. In the presence of the two salts of Au (III) and Ag (I), the use of sodium citrate led to aggregation and precipitation of the particles within 5 h. The use of PAA in the radiolytic fabrication of monometallic Au samples led to lower metal loading and the formation of larger non-homogeneous particles, whereas, for Au-Ag bimetallic structures, the higher PAA concentration allowed a greater amount of Ag to be deposited. In addition, the catalytic activity of PAA-stabilized Au remained higher than that of Cit-stabilized Au samples in view of the mass efficiency. Nevertheless, further optimization of the synthesis conditions (dose rate, pH, and nature of stabilizing agents) is necessary to acquire smaller NP sizes and a homogeneous distribution.

During the synthesis, the galvanic replacement was driven by the difference in redox potentials. Initially reduced Ag atoms re-oxidized back to Ag (I) ions by transferring the electrons to Au (III) ions to form metallic gold. Such phenomenon led to a bimetallic composition with minor Ag content that led to a beneficial electronic effect between Au and Ag atoms and allowed the onset potential to be lowered in the glycerol electro-oxidation reactions.

Our first results provide insights into the catalytic activity in glycerol electro-oxidation and its dependence on the electrode composition. The comparison via the normalization of the current density per unit mass (A·mg^−1^) showed that the maximum peak current density of monometallic and bimetallic electrodes (8 and 2.4 A·mg^−1^, respectively) was higher than some other published works (0.6–1 A·mg^−1^) [25,88].

The onset potential of 0.55 V vs. RHE of the Au-Ag bimetallic particles is related to reports in the literature. The materials exhibited good selectivity in glycerol electro-oxidation, whereby only two products of high interest, formate (49–64%) and glycolate (35–43%), were obtained at 1.14 V vs. RHE. Compared to the Au catalyst, the Au-Ag bimetallic specifically enhanced production of formate.

## Data Availability

Data are available upon request.

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
