# Peer review of "Radiolysis-Assisted Direct Growth of Gold-Based Electrocatalysts for Glycerol Oxidation"

_nanomaterials, 2023, doi:10.3390/nano13111713_

Round 1

Reviewer 1 Report

The paper describes a completely novel nanoparticle synthesis which, despite its relatively limited applicability (radioactive isotope required), could easily have a great future in catalytic applications of particles, as demonstrated by the authors in glycerol valorization tests. The article is well written with neat English, containing only a few small typos. The logical build-up and good articulation leads the reader well, after a small minor revision I recommend the article for publication.

1.     The material characterization part lacks the description of technical parameters of the X-ray diffraction measurements. It is also worth mentioning the XRD measurements in the abstract.

2.     The systematic presence of XRD reflections of unknown origin is very interesting and could be further investigated by infrared or Raman spectroscopy measurements to see if it could be related to the presence of some oxidized form.

3.     The authors have presented the literary background of the work in great detail, but, it would have been worth mentioning some examples of recent unconventional (like the radiolysis) metallic nanoparticle syntheses, for example in sonochemistry, mechanochemistry and microwave chemistry.

Sonochemical synthesis of Cu@Pt bimetallic nanoparticles, Molecules, 2022, 27, 5281.

Mechanochemically modified hydrazine reduction method for the synthesis of nickel nanoparticles and their catalytic activities in the Suzuki-Miyaura cross-coupling reaction, Reac. Kinet. Mech. Cat., 2019, 126, 857-868.

High-speed but not magic: Microwave-assisted synthesis of ultra-small silver nanoparticles, Langmuir, 2018, 34, 147-153.

4.     The authors have prepared manuscript in lengthy manner, which makes it a little difficult to understand the whole work, it might be worth moving some more analysis to the supporting information file. 

Author Response

Manuscript ID: nanomaterials-2386248

TITLE: Radiolysis-assisted direct growth of gold-based electrocatalysts for glycerol oxidation

Reply to Reviewers’ comments

We would like to thank the reviewers for the careful reading of our manuscript and for the valuable comments. Please find below our point-by-point responses to reviewers’ comments: the revised version of the manuscript was carefully examined and additional clarifications were added, highlighted in blue in the manuscript.

Reviewer 1

Comment 1: The material characterization part lacks the description of technical parameters of the X-ray diffraction measurements in the paragraph. It is also worth mentioning the XRD measurements in the abstract.

Response: The description of the technical parameters of the XRD analysis was added in the paragraph Materials and Methods. The XRD method was mentioned in the abstract.

Comment 2: The systematic presence of XRD reflections of unknown origin is very interesting and could be further investigated by infrared or Raman spectroscopy measurements to see if it could be related to the presence of some oxidized form.

Response: Indeed, the advice of additional IR or Raman studies to complement our work in understanding the structure and composition of surface particles is worthwhile. However, we were unable to perform such experiments (our samples are not suitable for direct IR analysis and some interferences arising from different species during Raman spectroscopy).

Comment 3: The authors have presented the literary background of the work in great detail, but, it would have been worth mentioning some examples of recent unconventional (like the radiolysis) metallic nanoparticle syntheses, for example in sonochemistry, mechanochemistry and microwave chemistry.

  • Sonochemical synthesis of Cu@Pt bimetallic nanoparticles, Molecules, 2022, 27, 5281.
  • Mechanochemically modified hydrazine reduction method for the synthesis of nickel nanoparticles and their catalytic activities in the Suzuki-Miyaura cross-coupling reaction, Reac. Kinet. Mech. Cat., 2019, 126, 857-868.
  • High-speed but not magic: Microwave-assisted synthesis of ultra-small silver nanoparticles, Langmuir, 2018, 34, 147-153.

Response: We appreciate your suggestions and add these references in the Introduction.

Comment 4: The authors have prepared manuscript in lengthy manner, which makes it a little difficult to understand the whole work, it might be worth moving some more analysis to the supporting information file.

Response: The suggestion has been taken into consideration. To lighten the main text, we moved some parts (e.g., XPS survey, Nyquist plots) from the main text to the supporting information materials. The figure numbering was corrected accordingly.

Reviewer 2 Report

The reviewed manuscript discusses a new method for the synthesis of mono- and bimetallic nanoparticles using radiolysis. Such parameters of nanoparticles as shape, size, elemental and chemical composition are determined. The features of the spatial distribution of particles on the substrate surface are established. Methods of controlling the characteristics of nanoparticles are revealed. In addition, the first experiments on the electrocatalytic decomposition of glycerol were carried out.

In general, the study was carried out at a high scientific level using methods adequate to the tasks. The reliability of the conclusions is beyond doubt. I believe that this manuscript will attract a wide range of readers.

Remarks:

1. It is advisable to give transcripts of abbreviations as they are introduced. See RHE for example.

2. It is desirable to distinguish between the designations of references “[]” and substances “[]”, lines 244 and 245 and further.

3. Is it possible to give the distribution of the nanoparticles obtained in different conditions by size?

4. Do the measured parameters of the systems (for example, the distribution of Au and Ag over nanoparticles' surface) depend on the size of the nanoparticles?

5. Does the glycerol decomposition products depend on the size of nanoparticles and their distribution over the CP surface?

Author Response

Manuscript ID: nanomaterials-2386248

TITLE: Radiolysis-assisted direct growth of gold-based electrocatalysts for glycerol oxidation

Reply to Reviewers’ comments

We would like to thank the reviewers for the careful reading of our manuscript and for the valuable comments. Please find below our point-by-point responses to reviewers’ comments: the revised version of the manuscript was carefully examined and additional clarifications were added, highlighted in blue in the manuscript.

Comment 1: It is advisable to give transcripts of abbreviations as they are introduced. See RHE for example.

Response: The comment was considered in the new version: the RHE full name was revealed, and other abbreviations were checked in the whole manuscript.

Comment 2: It is desirable to distinguish between the designations of references “[]” and substances “[]”, lines 244 and 245 and further.

Response: We have revised the manuscript to remove “[]” for concentrations.

Comment 3: Is it possible to give the distribution of the nanoparticles obtained in different conditions by size?

Response: The suggestion is highly appreciated. Indeed, knowing the effect of synthesis conditions and reaction mixture compositions onto the size of formed particles would be interesting. However, since this radiolytic synthesis route still requires further optimization, authors think that estimation of the size distribution is not worthy at this stage. Determination of the size distribution and mean size is planned for the future studies which will involve optimization of the synthesis procedure.

Comment 4: Do the measured parameters of the systems (for example, the distribution of Au and Ag over nanoparticles' surface) depend on the size of the nanoparticles?

Response: The performance of the electrode depends on the size of the deposited particles. However, many other factors influence the measured parameters such as mass charge, surface density, crystal structure and orientation. However, it is not possible at this stage to establish a direct link between the particle size and the electrode parameters, as the other morphological characteristics mentioned require further optimization. Future studies plan to optimize the synthesis conditions to better control these characteristics.

Comment 5: Does the glycerol decomposition products depend on the size of nanoparticles and their distribution over the CP surface?

Response: The answer is similar to the comment 4.

Reviewer 3 Report

This paper describes gold-based electrocatalysts for glycerol oxidation prepared by radiolysis, which is targeted to hydrogen production. Electrodes are analyzed in detail. Unfortunately, hydrogen production experiment is not conducted in this paper, which makes difficult to evaluate this work. Comparison of the present electrocatalysts with conventional ones is not well discussed with regard to advantage of use of radiolysis. The resulted electrochemical data do not seem to be exceptionally better than conventional data such as shown in 29 and 89. Implication of this work is not clear, and this paper cannot be recommended for publication in Nanomaterials in its present form.

Author Response

We would like to thank the reviewers for the careful reading of our manuscript and for the valuable comments. Please find below our point-by-point responses to reviewers’ comments: the revised version of the manuscript was carefully examined and additional clarifications were added, highlighted in blue in the manuscript.

Comment 1: This paper describes gold-based electrocatalysts for glycerol oxidation prepared by radiolysis, which is targeted to hydrogen production. Electrodes are analyzed in detail. Unfortunately, hydrogen production experiment is not conducted in this paper, which makes difficult to evaluate this work. Comparison of the present electrocatalysts with conventional ones is not well discussed with regard to advantage of use of radiolysis. The resulted electrochemical data do not seem to be exceptionally better than conventional data such as shown in 29 and 89. Implication of this work is not clear, and this paper cannot be recommended for publication in Nanomaterials in its present form.

Response: The objective of this paper is to explore the proof-of-concept of direct radiolysis-assisted growth of nanostructured gold and gold-silver particles on a gas diffusion electrode to produce self-contained electrocatalysts for the selective electrooxidation of glycerol. To our knowledge, this is the first time that the radiolytic reduction of metal salts on a 3D carbon substrate is described. As we mentioned in the manuscript, other unconventional synthesis routes were reported such as microwave-assisted, sono- and mechanochemical procedures, but they have not attempted to deposit metal particles on a 3D substrate.

The electrochemical data obtained in this paper have been compared with the literature work, although a fair comparison is not obvious due to studies that were not performed under the same experimental conditions and not always reported in sufficient detail. Our initial results clearly show that the maximum current density of our materials, via normalization of current density per unit mass, is between 2.4 and 8 A.mg-1, which is higher than recently published work (0.6-1 A.mg-1) [29], [91]. In addition, the initiation potential obtained in this work is relevant to the literature reports. The Au-Ag bimetallic particles synthesized in this work reduced the onset potential to 0.55 V vs. RHE, which could lead to a lower overpotential in the electrolysis cell.

Round 2

Reviewer 3 Report

My concern is that the advantages or interesting features of radiolysis appear not well described in this paper, although authors emphasize radiolysis in introduction. Is it possible to compare with microwave-assisted, sono- and mechanochemical procedures? Hydrogen generation is not conducted in this paper, but electrooxidation of glycerol is carried out. Accordingly, the discussions of this paper do not seem consistent. Detail explanations of Table 5 on onset potential, peak current density and selectivity may help, which include comparison of normalized current density per unit mass between 2.4 and 8 A.mg-1, which is higher than recently published work (0.6-1 A.mg-1).

Author Response

Cover letter of revision

Manuscript ID: nanomaterials-2386248

TITLE: Radiolysis-assisted direct growth of gold-based electrocatalysts for glycerol oxidation

Reply to Reviewer comments

We would like to thank the reviewer for its valuable comments. Please find below our point-by-point responses to reviewer comments: the revised version of the manuscript was carefully examined and additional clarifications were added, highlighted in red in the manuscript.

Reviewer 3

Comment 1: My concern is that the advantages or interesting features of radiolysis appear not well described in this paper, although authors emphasize radiolysis in introduction. Is it possible to compare with microwave-assisted, sono- and mechanochemical procedures?

In the introduction, the following paragraph:

“Previously, other unconventional synthesis routes were reported such as microwave-assisted, sono- and mechanochemical procedures, but they have not attempted to deposit metal particles on a 2D substrate.[44–46]”

The paragraph was removed and remplaced by:

 “Other unconventional physical synthesis routes exist, such as microwave-assisted, sono and mechanochemical procedures based on high temperatures and pressures, high-energy ball mills or low-temperature ultrasonic frequencies. These procedures are effective in producing metallic nanoparticles,[40]–[42] but, unlike radiolysis, they do not induce homogeneous reduction and nucleation throughout the sample volume, (solutions or heterogeneous media), leading to homogeneous nucleation and growth of NPs at room temperature. Also the high reducing power of solvated electrons enables reduction of salts of non-noble metals (such as Fe, Ni or Co), which are difficult to re-duce by chemical methods at room temperature. Another advantage of radiolysis is that it is a powerful method to synthesize bimetallic nanoparticles of controlled size and structure (core-shell or alloys).[18], [43]–[45]. To our knowledge, this is the first time that the radiolytic reduction of metal salts on a solid flat substrate is described.”

Comment 2: Hydrogen generation is not conducted in this paper, but electrooxidation of glycerol is carried out. Accordingly, the discussions of this paper do not seem consistent. Detail explanations of Table 5 on onset potential, peak current density and selectivity may help, which include comparison of normalized current density per unit mass between 2.4 and 8 A.mg-1, which is higher than recently published work (0.6-1 A.mg-1).”

As appropriately pointed out by the reviewer, this manuscript only presents the performance of selective electrooxidation of glycerol over gold-based electrocatalysts, and hydrogen generation is not conducted in this work. In this sense, the Introduction and Conclusion have been rewritten.

  • In introduction, the following paragraph was removed:

“One of the sustainable routes to global decarbonization is the efficient production of molecular hydrogen that produces clean energy as compared to fossil fuels. Nowa-days, more than 95% of hydrogen is produced by steam reforming of methane, re-forming of oil and gasification of coal, which results in enormous carbon emissions.[1, 2][1, 2] The production of hydrogen by electrolysis of water does not produce CO2 emissions, but requires a significant energy input (cell voltage of 1.8-2.0 V).[2-4][2-4] However, numerous studies have shown that the energy input of electrolysis leading to hydrogen co-production can be significantly lowered by replacing the reaction on the anode (oxygen evolution reaction, OER) with the oxidation of alcohols, resulting in a cell potential of the electrolysis process below 1 V.[5-6][5-6] Glycerol is a promising alcohol precursor for several reasons. Glycerol production as a by-product in the bio-diesel industry has been increasing since the early 2000s, which on the one hand lowers its market price, and on the other hand encourages its use as a feedstock. Moreover, the selective electro-oxidation of glycerol can lead to value-added products such as glyceric acid, dihydroxyacetone, tartronic acid, glycolic acid, formic acid, etc., the commercialization of which leads to a decrease in the overall cost of the H2 coproduct.[7]”

This paragraph was replaced by the following paragraph:

“Selective electro-oxidation of glycerol has been proposed as the most viable pathway for the production of value-added chemicals from waste biodiesel waste, as well as a cogeneration pathway for the production of green H2 using high efficiency electrolyzers.[1]–[3] Glycerol production has been increasing since the early 2000s, which on the one hand lowers its market price, and on the other hand encourages its use as a feedstock. The selective electro-oxidation of glycerol can lead to value-added products such as glyceric acid, dihydroxyacetone, tartronic acid, glycolic acid, formic acid with applications in many fields.[4], [5]“

  • In Conclusion, the following paragraph was removed:

“Besides, the onset potential achieved in this work is relevant with reports in the literature. The Au-Ag bimetallic particles synthesized in this work afforded to decrease the onset potential to 0.55 V vs RHE, which might lead to a lower overpotential in the electrolysis cell. This opens new perspectives in the generation of H2 by electrolysis of alcohols rather than by direct electrolysis of water at a much higher potential (requiring 1.8-2.0 V vs RHE).”

This paragraph was replaced by the following paragraph:

“ The onset potential of 0.55 V vs RHE of the Au-Ag bimetallic particles is relevant with reports in the literature. The materials exhibited good selectivity in glycerol elec-trooxidation, whereby only two products of high interest, formate (49-64%) and glyco-late (35-43%), have been obtained at 1.14 V vs RHE. Compared to Au catalyst, the Au-Ag bimetallic specifically enhanced production of formate.”

  • Concerning Table 5, the following discussion was inserted in the text.

The results of this work were compared to other literature precedents in Table 5. Although it is difficult to make direct comparison with other works (due to different conditions of the synthesis, metal loading and measurement), values of onset potential (Eonset), peak current density (jp) and selectivity can give general understanding of the relative performance. Our tests were performed at a much lower glycerol concentration of 0.1M. As the glycerol concentration directly influences the measured current, normalization of jp values as a function of glycerol concentration would highlight the very good performance of the bimetallic electrocatalysts prepared by radiolysis. In addition, similar to that observed in [22], [89], [90], bimetallic particles contributes to de-crease the onset potential of glycerol oxidation to lower values. Selectivity of mono-metallic towards the formation of the primary alcohol glyceric acid was dominant for Au supported on different supports [86], [88] (at +1.9V and +1.6 V vs RHE, respectively) and could be explained by weaker interaction between the nanoparticles and glyceric acid, allowing its desorption and hindering its further decomposition into two and one carbon products. For direct radiolysis-assisted growth of monometallic and bimetallic gold-silver nanostructured particles, performed at a lower potential of +1.14 V vs RHE, the interaction between the particles and the support could contribute to a different selectivity of glycerol electrooxidation towards a significant cleavage of the C-C bond and the main production of two- and one-carbon products.